# Detection of anomalies in the UV/Vis reflectances from the Ozone Monitoring Instrument

Nick Gorkavyi[1], Zachary Fasnacht[1], David Haffner[1], Sergey Marchenko[1], Joanna Joiner[2], Alexander Vasilkov[1]

[1] Science Systems and Applications, Lanham, MD, USA
[2] National Aeronautics and Space Administration (NASA), Goddard Space Flight Center (GSFC), Greenbelt, MD, USA

**Correspondence**: Nick Gorkavyi (nick.gorkavyi@ssaihq.com)

**Abstract.** Various instrumental or geophysical artifacts, such as saturation, stray light, or obstruction of light (either coming from the instrument or related to solar eclipses), negatively impact satellite measured ultraviolet and visible Earthshine radiance spectra and downstream retrievals of atmospheric and surface properties derived from these spectra. In addition, excessive noise such as from cosmic ray impacts, prevalent within the South Atlantic Anomaly, can also degrade satellite radiance measurements. Saturation specifically pertains to observations of very bright surfaces such as sun glint over open water or thick clouds. When saturation occurs, additional photoelectric charge generated at the saturated pixel may overflow to pixels adjacent to a saturated area and be reflected as a distorted image in the final sensor output. When these effects cannot be corrected to an acceptable level for science quality retrievals, flagging of the affected pixels is indicated. Here, we introduce a straightforward detection method that is based on the correlation, *r,* between the observed Earthshine radiance and solar irradiance spectra over a 10 nm-spectral range; our Decorrelation Index (DI for brevity) is simply defined as DI=1-*r*. DI increases with anomalous additive effects or excessive noise in either radiances, the most likely cause in data from the Ozone Monitoring Instrument (OMI), or irradiances. DI is relatively straight-forward to use and interpret and can be computed for different wavelength intervals. We developed a set of DIs for two spectral channels of the OMI, a hyperspectral pushbroom imaging spectrometer. For each OMI spatial measurement, we define 14 wavelength-dependent DIs within the OMI visible channel (350-498 nm) and 6 DIs in its ultraviolet 2 (UV2) channel (310-370 nm). As defined, DIs reflect a continuous range of deviations of observed spectra from the reference irradiance spectrum that are complementary to the binary Saturation Possibility Warning (SPW) flags currently provided for each individual spectral/spatial pixel in the OMI radiance data set. Smaller values of DI are also caused by a number of geophysical factors; this allows one to obtain interesting physical results on the global distribution of spectral variations.

## 1 Introduction

The Ozone Monitoring Instrument (OMI) is a Dutch/Finnish ultraviolet (UV) and visible (Vis) wavelength spectrometer that is on board NASA's Aura satellite launched on July 15, 2004. It has provided one to two day global coverage for several important atmospheric trace gases including ozone ($O_3$), sulphur dioxide ($SO_2$), nitrogen dioxide ($NO_2$), and formaldehyde (HCHO) as well as information about clouds and aerosols (Levelt et al., 2018). OMI has contributed to studies of atmospheric pollution, climate-related agents, and stratospheric chemistry (Levelt et al., 2018,), lead to the first observation of glyoxal ($C_2H_2O_2$) from space (Chan Miller et al., 2014), and provided precise long-term records of solar spectral

irradiances (Marchenko and Deland, 2014). OMI's data have contributed to medium-range weather and air quality forecasts, as well as to detection and tracking of volcanic plumes (Hassinen et al., 2008; Krotkov et al., 2015; Levelt et al., 2018). OMI measurements also provide estimates of tropospheric ozone columns (e.g., Sellitto et al., 2011; Ziemke et al., 2017). Several similar sensors are currently in orbit, including the Tropospheric Monitoring Instrument (TROPOMI) on board the Copernicus Sentinel-5 precursor (S5P) satellite, Ozone Mapping and Profiler Suite/Nadir Mapper (OMPS/NM) on Suomi

NPP and NOAA-20, and Global Ozone Monitoring Experiment 2 (GOME-2) instruments on European Organisation for the Exploitation of Meteorological Satellites (EUMETSAT) MetOp platforms.

      Non-linear effects can impact the measured signal when not properly corrected. This can degrade Earthshine radiance measurements from passive solar backscatter UV/Vis satellite spectra and thus impact retrievals of atmospheric constituents

and surface properties. There are several potential sources for these effects. Saturation occurs when bright light causes the number of electrons in a sensor pixel to exceed either the maximum charge capacity of an individual charge-coupled device (CCD) photodiode, or the maximum charge transfer capacity of the sensor. Blooming and other artifacts related to charge transfer on the CCD may also affect the quality of the measured spectrum when electrons from a saturated pixel overflow to a neighboring pixel, causing distortion of its signal and frequently rendering affected data useless. Charge transfer and

readout errors can also result in a distorted spectra, as can be the case with an error correction for detector smear. Hereafter we refer to the spatial domain of the two-dimensional CCD as rows (30 or 60 simultaneously acquired scenes), and the spectral domain as columns. Per OMI design, the CCD readout is in  spatial pixels more easily, and therefore the blooming or charge readout-related effects are expected to predominantly occur between different spatial rows.

Retrievals of atmospheric gases or aerosols can be compromised when observing very bright surfaces such as sun glint in low wind speed conditions (Cox and Munk, 1964; Kay et al., 2009; Butz et al., 2013; Feng et al., 2016), as well as over scenes predominantly covered by optically thick clouds.  Saturation caused by Sun glint routinely occurs in the visible imagery of the MODerate Resolution Imaging Spectroradiometer (MODIS) flying on NASA's Aqua and Terra satellites. MODIS data show a gradual increase of saturated data towards the red and NIR bands, reaching around 1500 pixels, or ~

0.03% of pixels, in a granule at 869 nm (Singh and Shanmugam, 2014). The Orbiting Carbon Observatory-2 (OCO-2) and similar greenhouse gas monitoring instruments occasionally point directly at the sun-glint. The OCO-2 in-orbit checkout activities revealed an unexpectedly high signal from Lake Maracaibo, Venezuela on August 7, 2014. This signal saturated all 3 channels and was attributed to an oil slick on a wave-free lake. After this event, known as the Lake Maracaibo Saturation Incident, an automated saturation warning algorithm was incorporated into the OCO-2 processing to identify such events

(Crisp et al., 2017). Solar glint from ocean and clouds, as well as "saturation tails" or blooming effects are seen in many images from the Earth Polychromatic Imaging Camera on the Deep Space Climate Observatory (EPIC/DSCOVR) (Varnai et al., 2019).  TROPOMI also experiences detector saturation and blooming problems, typically caused by bright tropical clouds seen in bands 4 (400-499 nm) and 6 (725-786 nm). Bands 7 (2300-2343 nm) and 8 (2342-2389 nm) mostly react

negatively to sun glint. Currently, blooming areas are not detected by the TROPOMI L0-1b processor. A flagging algorithm
is under development (Rozemeijer and Kleipool, 2019; Ludewig et al., 2019).

A set of 16 operational flags, called the Saturation_Possibility_Warning (SPW) flags are currently included in the OMI level
1b data set. SPWs are designed to flag OMI pixels with 16 various radiation anomalies (e.g., saturation, stray light,
nonlinearity). These flags are defined for each OMI wavelength: 751 wavelengths of the Vis spectrum and 557 wavelengths
of the UV2 spectrum (GDPS, 2006). All of the 16 SPW flags are binary; a pixel with any degree of abnormality (e.g.,
saturation) at a given wavelength is marked as possibly bad.

Here, we describe a new approach to identify potentially erroneous OMI data based on the correlation $r$ between the
observed back-scattered Earthshine spectrum and a reference solar spectrum computed over limited spectral regions.
Earthshine spectra differ from the solar spectra due to Rayleigh, rotational-Raman, aerosol and surface scattering as well as
absorption of radiation by ozone and other atmospheric components. Most of these factors, with the exception of strong
ozone absorption in the UV, amount to secondary effects on the correlation coefficient between the solar and Earthshine
spectra within a limited spectral window. Under normal conditions (lack of detectable instrument-imposed spectral
distortions) and for a reasonably narrow (5-10 nm, for practical purposes, with a moderate-resolution spectral instrument)
spectral window, the degree of correlation depends mainly (but not exclusively) on the number and strength (depth) of solar
Fraunhofer features, once we take into consideration additional factors (differences in spectral resolution, finite signal-to-
noise of measurements, mis-alignment of the wavelength grids, among others) that tend to degrade the correlation. In the
windows with well-defined solar absorption spectral features, the correlation coefficient may gradually approach unity for
the scenes acquired with S/N>>100 – a condition met in a majority of OMI UV2-Vis reflectance spectra.  Assuming the
radiances and irradiances have the same spectral resolution, comparable S/N, and are closely co-aligned in the wavelength
domain, the correlation coefficient between the earthshine and the solar 'etalon' (assumed to be distortion-free)   should be
highly sensitive to any distortions  in the former,  leading to  rapidly decreasing correlation in saturated scenes (solar glint or
bright clouds) or under other anomalous conditions, such as cosmic ray hits on the detector.

We apply our approach to OMI data and analyze individual cases and global distributions of flagged data. While these
effects have been known for some time and dealt with, to some extent, the prevalence of the different effects globally for a
particular instrument has rarely been documented. This work provides a detailed analysis of spectrum-distorting effects in
the  OMI case, as well as a general and straight-forward approach that may be applied to similar instruments (TROPOMI,
OMPS, GOME-2, etc.) to identify and filter out suspect or erroneous data.

## 2 Data and Methods

### 2.1 The Ozone Monitoring Instrument

The Aura satellite that hosts OMI is in a polar Sun-synchronous orbit with a local equator crossing time of 13:45. OMI is a nadir-looking, push-broom UV/Vis grating spectrometer (Levelt et al., 2018). The light entering the telescope is depolarized using a scrambler and then split into two channels: the UV (wavelength range 264–383 nm) and the Vis (wavelength range 349–504 nm: Dobber et al., 2006; Schenkeveld et al., 2017). The UV channel is further divided into the two sub-channels, UV1 (264-311 nm, 0.63 nm resolution and 0.21 nm sampling) and UV2 (307-383 nm range, 0.42 nm resolution with 0.14 nm sampling). Measurements are collected on two-dimensional charge-coupled device (CCD) sensors used for the UV and Vis channels. Spectral information is dispersed along one dimension of each CCD and spatial is imaged on the other. Each channel has a devoted frame-transfer CCD detector with 6e5 electrons/pixel full-well capacity. To avoid blooming and ellipsoid effects, the pixel filling should be kept below 3e5 electrons (Dobber et al., 2006). OMI also measures the solar irradiance once per day through the solar port. Here, we use the UV2 sub-channel and Vis channel only; in the UV1 channel, strong, variable ozone absorption renders our approach impractical.

In the global mode, each orbit spans the pole-to-pole sunlit portion, typically comprising 1644 along-orbit exposures, referred to as iTimes hereafter. The $114°$ viewing angle of the telescope corresponds to a 2600 km wide swath on the Earth's surface and consists of 60 simultaneously acquired rows or ground pixels across the track. In this mode, the OMI pixel size is $13 \times 24$ km$^2$ at nadir. The in-flight performance of OMI is discussed in Schenkeveld et al. (2017). The radiometric degradation of the OMI radiances since launch ranges from ~2 % in the UV channel to ~0.5 % in the Vis channel, which is much lower than any similar sensor (Levelt et al., 2018). One major anomaly has occurred with OMI, the so-called row anomaly (Schenkeveld et al., 2017); it is presumably caused by a partial detachment of insulation material exterior to the instrument and produces a number of anomalous effects on sun-normalized radiances. The row anomaly is discussed in detail in Section 3.4.

### 2.2 The Decorrelation Index (DI)

We introduce a new parameter, the decorrelation index (DI), and defined as $1 - r$, where $r$ is the Pearson correlation coefficient:

$$DI = 1 - r = 1 - \frac{\sum_{i=1}^{n}(x_i - \bar{x})(y_i - \bar{y})}{\sqrt{\sum_{i=1}^{n}(x_i - \bar{x})^2}\sqrt{\sum_{i=1}^{n}(y_i - \bar{y})^2}} \qquad (1)$$

with $\bar{x}$ (same for $\bar{y}$)

$$\bar{x} = \frac{1}{n}\sum_{i=1}^{n}x_i \qquad (2)$$

In (1) and (2) $x_i$ and $y_i$ are the individual sample points for radiance $I$ and irradiance $F_0$, respectively. DI is derived for radiances and irradiances at each spectral region: for OMI, 14 regions of ~10 nm ($n$= 51 wavelengths for each spectral region) in the Vis channel and 6 regions of ~10 nm ($n = 69$ wavelengths for each region) in UV2. For the standard solar spectrum or reference irradiance, we take an average of all solar spectra obtained by OMI in 2005. Each earthshine spectrum is re-gridded via linear interpolation to match the wavelengths of the averaged irradiance spectrum. An exact match between the radiance and irradiance spectral features gives DI = 0, whereas when the features in the radiance and irradiance spectra deviate, the DI approaches 1 to 2, where values greater than 1 indicate that irradiance and radiance spectra exhibit anti-correlation. Hence, cases of DI > 0 may indicate distortions of atmospheric spectra. Evidently $DI = 0$ for the simple case of a perfect match with $I = const * F_0$; if $I = -const * F_0$, then $DI = 2$. If $I$ and $F_0$ are completely unrelated, then $DI = 1$. Considering the 'smooth' (low-frequency) component of $I$ and $F_0$, we expect them to be generally correlated in the spectral regions relatively free of major atmospheric absorptions (ozone in particular). The correlation would be inevitably diminished by the wavelength-dependent Rayleigh scattering and surface reflectivity. Once a multitude of deep spectral lines is superimposed on a smooth envelope, DI will depend mainly on a match between the shape and position of these $I$ and $F_0$ spectral transitions, with the correlation depending on the S/N of the tested radiances and irradiances, and even more so on slight (in OMI's case) wavelength mis-alignments between radiances and irradiances, with the steep line flanks magnifying the differences.

An additional de-correlating factor is brought forth by the omni-present atmospheric rotational Raman scattering (e.g., Joiner et al., 1995). Under the circumstances, one may never expect $DI = 0$ save the exceedingly rare cases of a perfect solar glint. It is known that Pearson's correlation coefficient is sensitive to outliers, thus simplifying detection of spectral distortions in the high-resolution data compared to the low-resolution cases, with the latter tending to lessen the impact of additive components (the shallower lines are potentially less susceptible to stray light), as well as the wavelength mis-alignment (spectral blending of multiple features leading to partial canceling of distortions in the adjacent features). At a given spectral resolution and S/N, DI sensitivity may grow with increasing numbers and contrasts (depths) of spectral features in the chosen spectral window. At the same time, the DI is expected to be sensitive to artifacts associated with cosmic ray hits. The interval 440-480 nm, where there are few deep spectral lines, should be especially sensitive to geophysical factors, for example, to the wavelength-dependent albedo of the earth's surface. Note that there are cases when direct solar radiation $F_0$ is mixed with $I$ due to instrument problems (see below). Under this specific circumstance DI will decrease, since correlation between $(I + \delta F_0)$ and $F_0$ is always higher (thus DI lower) than between $I$ and $F_0$. A similar effect occurs with sunglint from the water surface, when the proportion of directly reflected sunlight in $I$ increases significantly. Note that in the current approach we do not compensate for the relatively smooth spectral differences imposed by atmospheric (Rayleigh scattering) and surface (wavelength-dependent albedo) factors, leaving this to the next DI version. This step would make DI more sensitive to the instrument-imposed anomalies, further disentangling those from the geophysical factors (see below).

In this initial version of the OMI DI, we use the spectral range 309.9-370.0 nm for UV2 and 349.9-498.4 nm for Vis. Overlapping of these ranges is useful for assessing the calibration between the UV2 and VIS channels. For solar zenith angles (SZA) > 90°, the radiance level drops, noise begins to dominate, and the DI grows rapidly. Therefore, we avoid SZA > 90° cases. The DI is sensitive to the degree of distortion of the reflectance spectrum, regardless of the cause of the distortion (saturation, crosstalk, noise etc), so that it detects distortions other than saturation. For example, the DI may detect electronic cross-talk (or blooming) effects in pixels adjacent to the saturated area. In a number of cases, the DI proves to be either more or less sensitive than the current SPW (Saturation_Possibility_Warning) flags reported in the OMI PixelQualityFlags filed of the Level 1b data, as shown in the next section.

The DI provides a range of values that describes the deviation of observed spectra from the reference irradiance spectrum, while the SPW flag is a binary value. The DI therefore allows flexibility in setting detection thresholds for damaged spectra for different applications. The DI value for a given spectral interval depends strongly on the number of Fraunhofer lines as well as presence of strong ozone absorption features within the wavelength range. Therefore, the DI values corresponding to likely damaged spectra vary somewhat for each spectral region. For example, the 14 DI divisions of the Vis spectrum generally fall into two distinct groups; for the first group, the value of DI above 0.01-0.03 is a sign of a significant distortion of the spectrum, while for the second group a typical distortion threshold value is larger (~0.1-0.4). The provisional (the user may redefine the values using the auxiliary data provided in the OMI DI product) DI thresholds were determined as follows. We used all available, mission-long OMI UV2 and Vis radiances. For each orbit and for every spectral window we constructed DI histograms. Then we selected numerous cases sampling the tails of the DI histograms. On a case-by-case basis, for different scenes and spectral windows, we found empirically the lowest-DI values that repeatedly separate the scenes with apparently normal (spectrally smooth, with the fine-structure, low-amplitude Raman-scattering features) and distorted reflectances. These DI thresholds approximately correspond to 99.995-99.998 percentiles in the DI distributions. We plan to provide a statistically rigorous threshold definition in the improved DI version.

Table 1 summarizes the DI wavelength bands and suggested threshold values corresponding to damaged spectra. These critical values should be treated as indicative. A user may define different thresholds depending on their application. We chose row 20 to determine these critical values.

**Table 1.** Chosen OMI DI spectral intervals and indicative DI thresholds for damaged spectra.

| Interval (UV2) | Wavelengths (nm) (for row 20) | Value DI as signature of distortion of spectra | Comments |
|---|---|---|---|
| 1 | 309.94-320.61 | - * | Strong ozone effects |
| 2 | 320.76-331.08 | >0.20-0.25 | Ozone effects |
| 3 | 331.23-341.24 | >0.35-0.45 | Weak ozone effects |

| | | | |
|---|---|---|---|
| 4 | 341.39-351.11 | >0.02-0.03 | Strong spectral lines |
| 5 | 351.25-360.70 | >0.02 | Strong spectral lines |
| 6 | 360.84-370.02 | >0.01 | Strong spectral lines |
| (Vis) | | | |
| 1 | 349.93-360.33 | >0.03 | Strong spectral lines |
| 2 | 360.54-370.93 | >0.01 | Strong spectral lines |
| 3 | 371.14-381.52 | >0.02 | Strong spectral lines |
| 4 | 381.73-392.11 | >0.01 | Strong spectral lines |
| 5 | 392.32-402.70 | >0.01 | Strong spectral lines |
| 6 | 402.91-413.29 | >0.06-0.08 | |
| 7 | 413.50-423.89 | >0.1-0.15 | |
| 8 | 424.10-434.50 | >0.02-0.03 | Strong spectral lines |
| 9 | 434.71-445.12 | >0.05-0.1 | |
| 10 | 445.32-455.74 | >0.25 | |
| 11 | 455.95-466.39 | >0.4 | |
| 12 | 466.60-477.05 | >0.4 | |
| 13 | 477.26-487.72 | >0.03 | Strong spectral lines |
| 14 | 487.93-498.41 | >0.2 | |

195   *The threshold depends on the row number.

The dependence of the threshold DI values on cross-track position is relatively minor, except for the first UV2 interval. For this interval, other cross-track position may carry different values, primarily due to ozone absorption (increasing towards the swath edges).  The DI thresholds depend on spectral resolution and S/N of the reflectances, hence the indicative values from

200   Table 1 may vary for different instruments.

## 3 Results

To study the DI, we first concentrate on scenes that are most likely to contain saturation effects: sun glint areas with relatively calm water surfaces and contiguous bands of deep convective clouds. Next, we examine the global DI distribution, which reveals other effects that damage observed spectra. We then investigate the impact of the row anomaly on the DI.

205   **3.1 Saturation over clouds**

A typical problematic cluster of bright clouds in the Pacific Ocean is shown in Fig.1a, where two zones are highlighted, a small northern zone (denoted A) and a large southern zone (marked as B). Figure 1c shows the number of wavelengths for a given pixel marked with the SPW flag as saturated. Figure 1b -1d shows the corresponding DI values for the Vis interval 414-424 nm. The DI indicates that the spectra in zone A are weakly affected, and in zone B they are badly damaged. Figure

210   1c shows the number of wavelengths for a given pixel marked with the SPW flag as saturated.

Figures 2 and 3 illustrate the properties of the DI that characterize the quality of a given part of the spectrum using a single parameter. Figure 2 shows an example of a spectrum with slight distortions that are captured by the SPW flags, but nevertheless has low values of the DI. Small deviations of the DI from 0 can result from geophysical effects, for example, an increased amount of ozone, and minor damage to the spectrum, as shown in Fig. 2. Those users who have strict requirements for the quality of the spectra should use the SPW flag in this case, which detects minor damage to the spectrum. Figure 3 shows the Vis spectrum for a pixel in zone B (indicated by an arrow in Fig. 1) corresponding to iTimes = 807, Row = 20. The radiance spectrum is saturated in the 400-465 nm range. In contrast with Fig. 2, damage in this spectrum is manifested in both the SPW flag and the DIs. The DIs reflects the degree of spectral damage, which in this case reaches a maximum near 450 nm. Based on the problem under study, a user can determine whether the spectrum is useful despite minor damage such as in zone A. In such cases, the SPW and DI may provide complementary information.

Reflectance on Figures 2 and 3 is defined as $\pi \cdot I / [F_0 \cdot \cos(\theta)]$ , where $I$ is the top-of-the-atmosphere (TOA) radiance, $F_0$ is the extraterrestrial solar flux, $\theta$ is the solar zenith angle (SZA). Usually, the wavelength dependence of TOA reflectance is fairly smooth, albeit with relatively low-amplitude, high-frequency structures due to rotational Raman scattering, also known as the Ring effect. Both zones in Fig.1 have high values of reflectance; for zone A, reflectance is between 0.95 and 1.0 (Fig. 2), while for zone B, reflectance is between 1.0 and 1.1 (Fig. 3). In some viewing directions the reflectance can exceed unity due to anisotropic angular distribution of the TOA radiance.

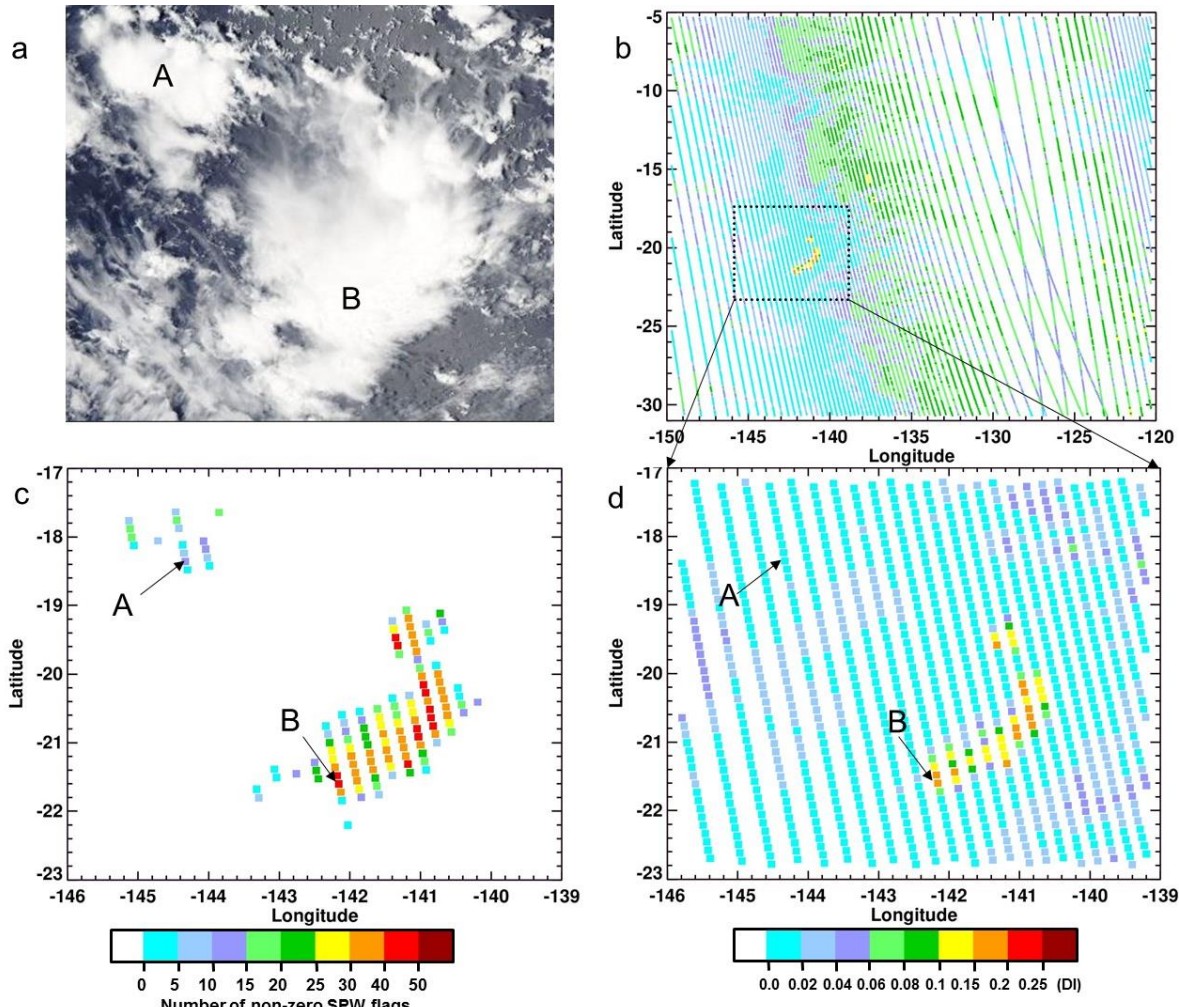

**Figure 1: a. Two cloud zones in the south Pacific on January 14, 2006 for orbit 7990: small northern zone labeled "A" and large southern labeled "B" (a) Aqua MODIS image; (b), (d) DI maps for the Vis spectral region 414-424 nm; (c) The number of wavelengths for a given pixel marked with the SPW (Saturation Possibility Warning) flag as saturated (the maximum number is 51 in this Vis spectral region).**

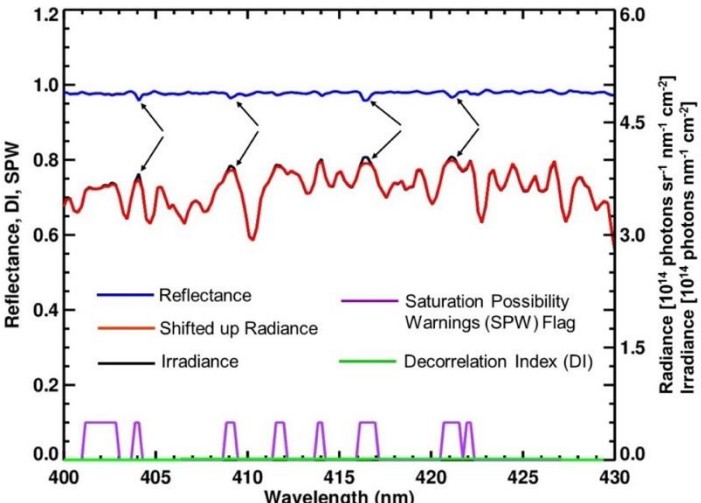

**Figure 2: Data for iTimes 839, row 15, orbit 7990, January 14, 2006 in zone A (this pixel is marked by arrows in Fig. 1(c,d). The blue line at the top of the picture is reflectance $\pi I /[F_0 \cos(\theta)]$, where $\theta$ is solar zenith angle. Reflectance in this zone has slight variations caused by minor saturation in the atmospheric spectrum as indicated by the arrows. The purple line shows the binary SPW (Saturation Possibility Warning) flags multiplied by 0.1. The green line is the DI<0.01 for bands 403-413, 413-424, 424-434 nm. The intensity of the radiance spectrum is shifted upwards slightly for clearer comparison with the irradiance spectrum.**

240

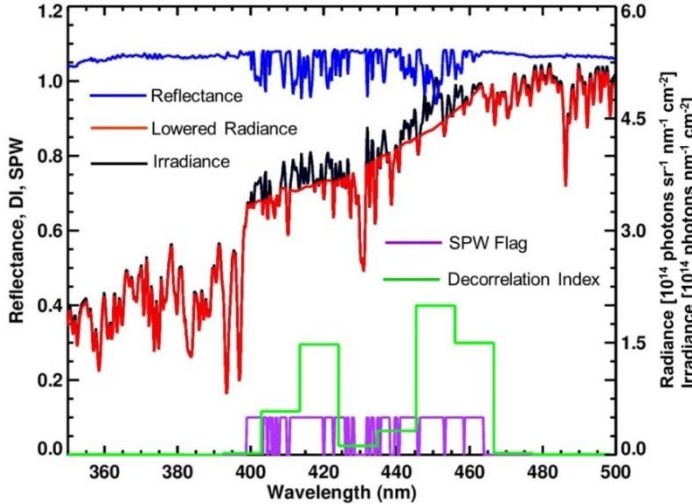

**Figure 3: Similar to Fig. 2 but for iTimes - 807, Row - 20, orbit 7990, January 14, 2006 in cloud region B (see arrows in Fig.1 c,d). The radiance was lowered by a few percent for better comparison.**

245

## 3.2 Saturation over lakes and oceans

The South American lake Salar de Uyuni is used for calibration of many satellite sensors (Lamparelli et al., 2003; Fricker et al., 2005). Salar de Uyuni is dry for most months of the year, but during the rainy season, it is filled with shallow water with strong direct reflectance from the sun. This may cause saturation of OMI's detectors. The lake, covered with shallow water, generated strong solar glint, for example, for orbit 7987 (January 14, 2006). Figure 4a shows this shallow lake on January 14, 2006 as observed by the Aqua MODIS sensor. The SPW flags (Fig. 4c) and DIs (Fig. 4b,d) for this case show that the lake generates two bright spots: southern and northern. The solar glint from the northern spot is so bright that the signal extends to nearby pixels (iTimes 823-825, Rows 11-14) and are detected by the DI (see also Cao et al., 2019; Shen et al., 2019 for examples of blooming in other sensors). The SPW flags is unset for significant portions of the affected OMI pixels (see Fig. 4).

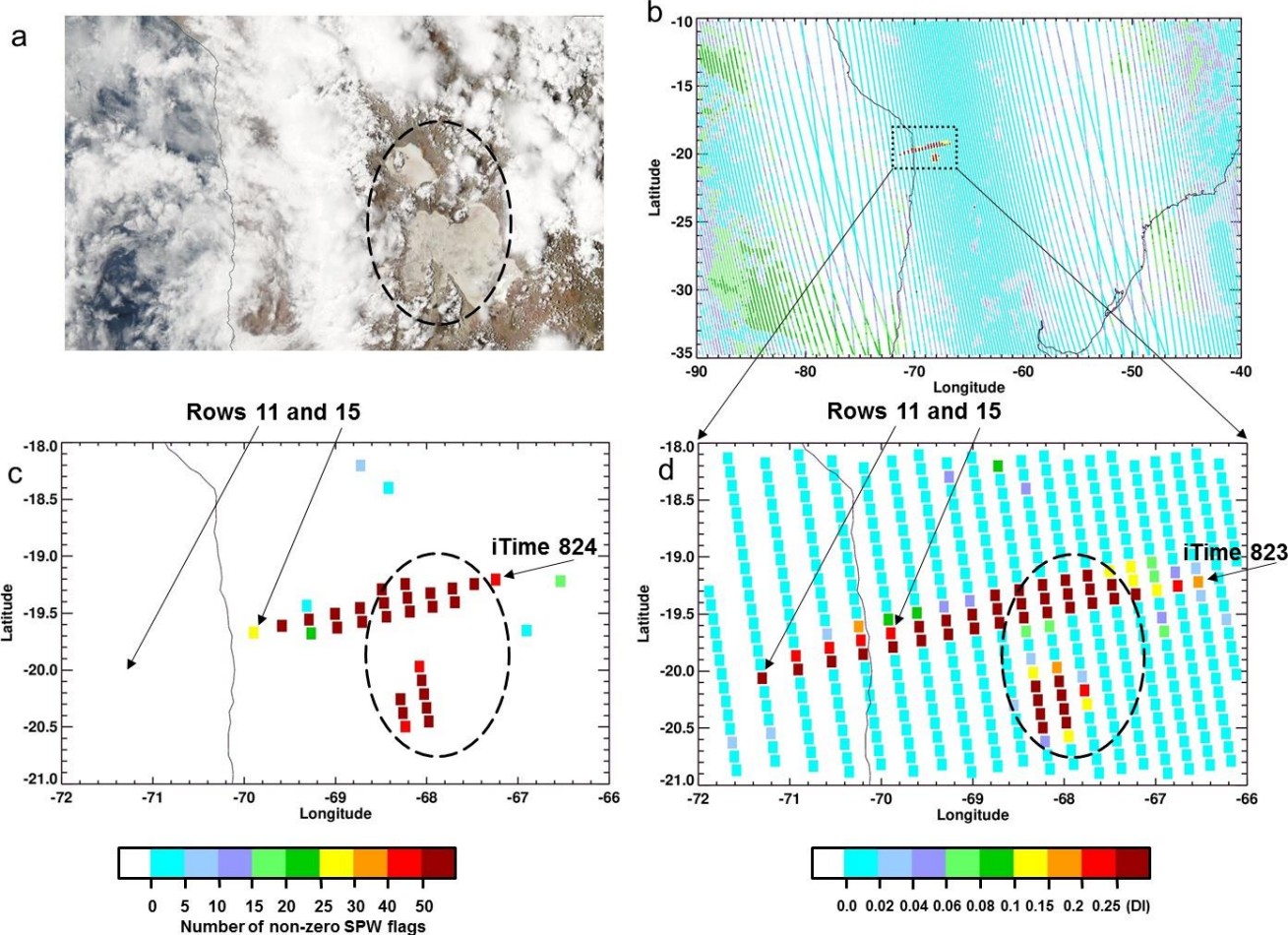

Figure 4: Similar to Fig. 1 but for an area near Salar de Uyuni, January 14, 2006, orbit 7987.

Figure 5 shows the Vis spectrum for a pixel where on the edge of the highly saturated region. This pixel may have superimposed effects of moderate saturation of the pixel itself as well as charge overflow from due to significant saturation in the neighboring pixels  (iTimes - 824, Row - 15). While the reflectance values of many of these pixels (rows ~11-15 in Figure 4) are in the expected  range (0.30.6) there are numerous cases where the final radiance signal is well beyond normal range, thus leading to high DIs.

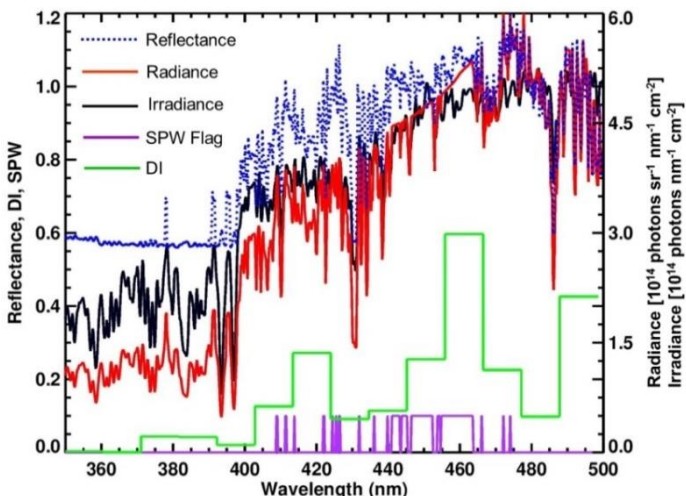

**Figure 5: Similar to Fig. 2 but for pixel iTimes 824, row 15, orbit 7987 indicated by arrows in Fig. 4(c,d) showing solar glint from Lake Salar de Uyuni.**

Figure 6 shows the spectrum of  a pixel where the prevailing  anomalous effect do not appear to be direct saturation (iTimes: 824, Row: 11). High DI values are seen for a number of corrupted parts of the spectrum where the SPW flags are zero. While the saturated case is straightforward to detect and interpret (e.g., the practically featureless radiances in Figures 3 and 4), these other spectral distortions may have a complex spectral envelope due to the differences in the wavelength sampling of the sequential OMI rows. The completely saturated spectral domains may trigger effects in the neighboring rows, indiscriminately affecting the involved wavelengths. However, in the case of a less severely saturated scene, there might be additional effects to consider. Per instrument design, the OMI wavelength grids form a 'spectral smile' in the row-wise direction. Inspecting the wavelength registration for a given CCD column (the spectral domain) while moving from the swath's edges towards nadir, one may notice gradual wavelength shifts between the adjacent rows. The wavelengths are increasing while moving from the edges to the center of the swath, thus forming a 'smile'. This may result in occasional augmented distortions around narrow, well-defined features in the spectral image in non-saturated pixels, while the signal for

other wavelengths in the spectrum may remain intact. Such occasional distortions could be mimicked and greatly outnumbered by a different effect that also stems from the spectral smile. In some cases, brightly lit (but not saturated) scenes border low-reflectance areas: e.g., the studied Salar de Uyuni case, cloud-front edges, or the edges of extended fresh

snow/ice fields. In these bordering low-reflectance scenes the greatly augmented spatial stray light could mimic a blooming effect caused by the spectral smile, thus leading to higher DI around strong, deep spectral transitions that may exceed the imposed threshold. In the OMI data sampling the high-contrast scenes, the spatial stray light effects induce wavelength shifts that affect trace-gas retrievals (Richter et al., 2020). Some of the above-threshold DIs in the global maps (mid-to-high latitudes, open-water scenes - see below) could be triggered by the high-contrast scenario.

290 .

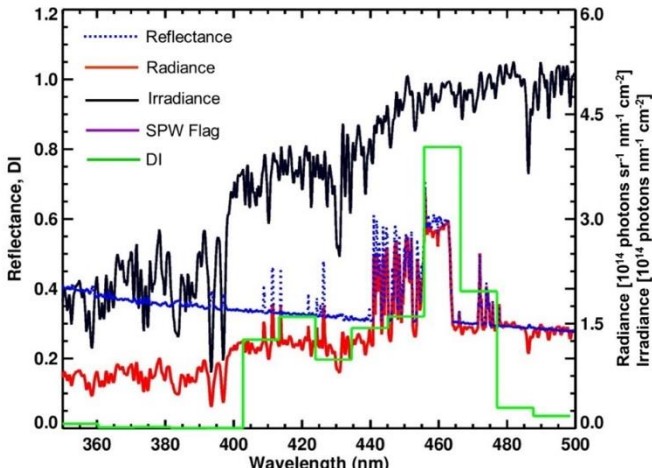

**Figure 6: Similar to Fig. 5 but for pixel iTimes 823, row 11, orbit 7987 (see arrows in Fig. 4d). SPW flags are zero for this case and**
**are not shown.**

An example of solar glint in the Caribbean Sea is shown in Fig. 7 for July 26, 2013. Effects of the glint for this case are detected in both the DI and SPW flags. Some of the pixels not marked by the SPW flags show high DI values that may be related to blooming or other effects associated with the impaired performance of neighboring pixels on the detector.


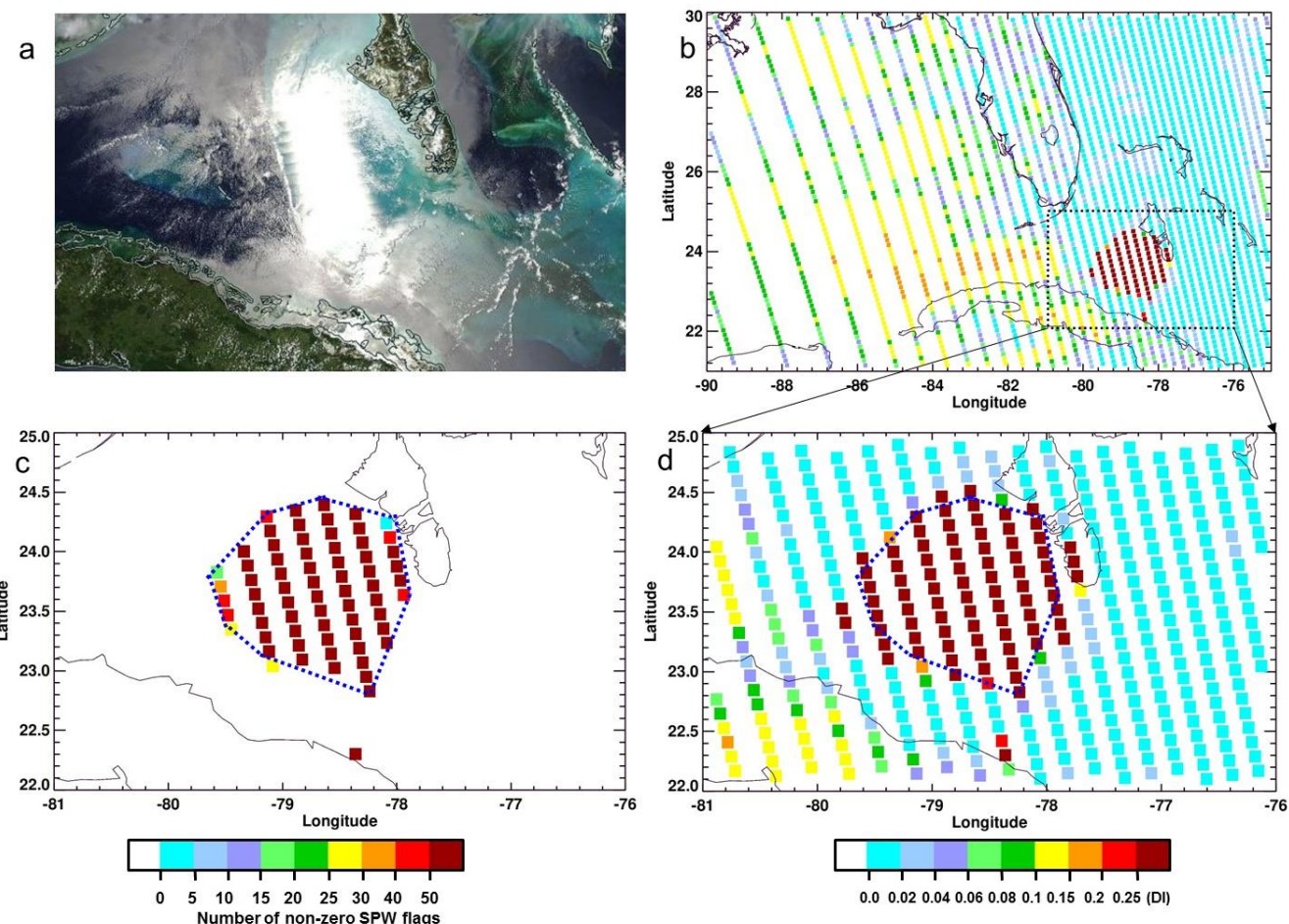

**Figure 7: Similar to Fig. 1 but for an area showing solar glint near the Bahamas (orbit 48034, July 26, 2013). Glint positions in (c) and (d) do not exactly match those in the RGB image (a). The area of pixels marked by the SPW flags are approximately delineated by the blue dotted line.**

### 3.3 Orbital and Global Distribution

Figures 8-10 show global distributions of the number of affected Vis spectra with DI > thresholds for spectra for March 2006 (Figs. 8 and 9) and for the entire 2006 (Fig. 10). The global DI behavior depends on many geophysical and instrumental processes. For example, Fig. 8a shows the spatial distribution of the number of spectra with DI > 0.03 for the 424.1-434.5 nm range. Figure 8b similarly shows distributions for DI > 0.25 in the 445.3-455.7 nm spectral window. Despite the different threshold DI values that characterize the distorted spectra, these two DIs show similar distributions of corrupted spectra associated with enhanced cosmic ray hits on the detectors within the South Atlantic Anomaly (SAA) region, glint, and a

solar eclipse zone. Though we cannot disentangle the contributing factors for the latter, here we mention two of them as the

likely causes of the high DI values (thus enhanced distortions in the reflectances): the low S/N of the eclipsed radiances, as

well as the drastically increased portion (compared to the normally-lit scenes) of the additive (straylight) component.

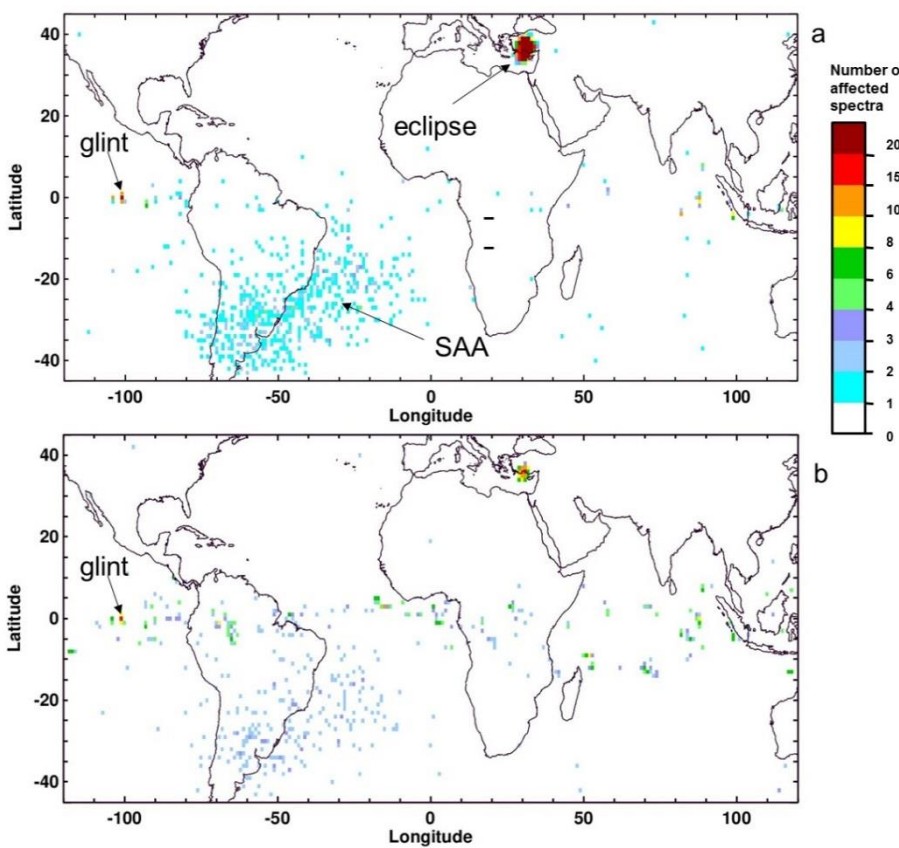

**Figure 8: Gridded (1°x1°) distributions of the number of affected spectra for March 2006; (a) Vis band 424.1-434.5 nm, DI > 0.03; (b) Vis 445.3-455.7 nm, DI > 0.25. The South Atlantic Anomaly (SAA) and a region affected by solar eclipse are clearly visible; the remaining pixels with high DI values are mostly associated with sun glints and bright clouds.**

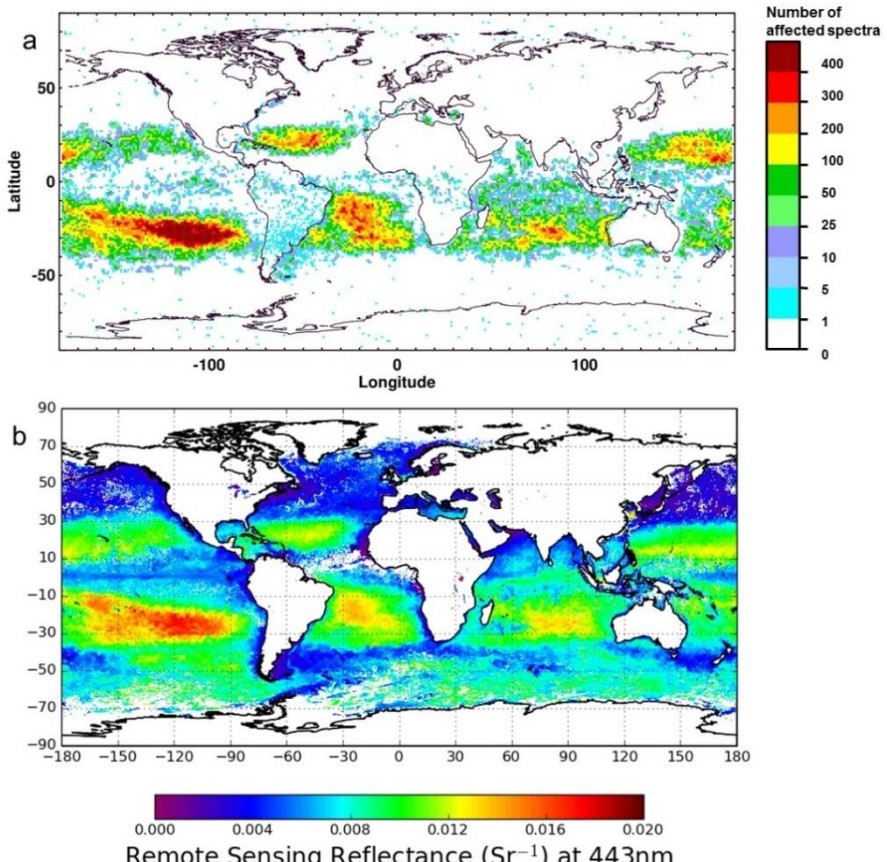

**Figure 9: (a) Gridded (1°x1°) distribution of a number of spectra with DI>0.1 for Vis 445.3-455.7 nm in March 2006; (b) ocean remote sensing reflectance for March 2006 at 443 nm from Aqua MODIS.**

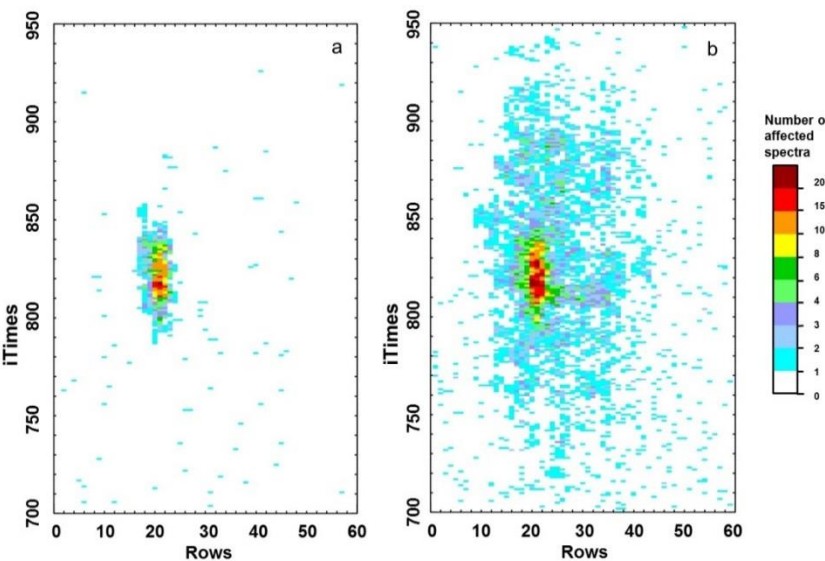

**Figure 10: Distribution of a number of affected spectra for 2006, Vis 445.3-455.7 nm; (a) DI > 0.6; (b) DI>0.25. Usually each pixel collects ~ 5000 spectra per year.**

The interpretation of low DI for normal spectra (for example, spectra with DI > 0.1 for Vis 445.3-455.7 nm) is quite complicated as low DI values depend on many factors. Figure 9 shows the spatial distribution of the number of spectra with DI > 0.1 in the 445.3-455.7 nm region compared with ocean reflectance at 443 nm. There is obvious spatial correlation between the spectra the DI identifies and ocean reflectance: larger numbers of such spectra correspond to ocean areas with higher reflectance. This is particularly pronounced in the southern Pacific Gyre whose waters exhibit extremely low bio-

productivity and thus are very bright in the blue region (Tedetti et al., 2007). The strong spectral dependence of water-leaving reflectance in the blue region in these extremely clear waters results in lower correlation with the solar spectrum. This may be attributed in part to vibrational Raman scattering that is prevalent in clear ocean waters (Vasilkov et al., 2002; Westberry et al., 2013). Additionally, the Pacific Gyre area is characterized by low cloudiness and low aerosol loadings. Therefore, in this area the relatively high proportion of the shown data comes from the surface, thus being more susceptible

to the Rayleigh and Raman scattering effects. These change the TOA radiances in different ways: the low-frequency spectral envelope is affected by Rayleigh, while the fine-scale structures are introduced by the vibrational and rotational Raman scattering. Both effects lead to 'distorted' reflectances, thus higher DIs.

Figure 10 a,b shows the orbital distributions of the 445.3-455.7 nm DI for DI > 0.25 and DI > 0.6 , respectively, plotted for

OMI detector rows (generally oriented east-west across the satellite track) versus iTimes (north-south orbital direction) for 2006. A block of 250 along-orbit exposures (iTimes) approximately covers $30^{\circ}$ in latitude. The middle of this band falls on the equator on March 22. During the year, this band shifts by $22.4^{\circ}$ to both the north and south. The zone around row 21 and iTimes 820 is an area of solar glint from the ocean surface (case DI > 0.6, Fig.10b) that does not change with season. The distribution of bright clouds with DI > 0.25 also shows a strong propensity for the geometrical conditions of solar glint

(Fig.10a). This is consistent with EPIC/DSCOVR data showing solar glint from clouds that contain oriented ice plates (Varnai et al., 2019). In the OMI case, the strongly saturated (or damaged) spectra with DI > 0.60 number about 2500 (~0.0005%) or ~7 spectra/day. Slightly affected spectra (0.25 < DI < 0.6) occur at a rate of ~0.002% or ~33 spectra/day.

### 3.4 The Row Anomaly

The row anomaly (RA) renders a significant portion of the OMI rows as unusable. The anomaly was clearly detected in two

rows in June 2007. In May 2008 the row anomaly spread to several other rows on the sensor. The row anomaly has continued to develop since then, with particularly swift changes around January 2009 and early fall of 2011. Currently about 33% of the UV2 rows are affected in the southern hemisphere parts of the OMI orbit. This increases to ~57 % in the northern hemisphere. These estimates are comparable in the Vis channels (Schenkeveld et al., 2017). Figure 11 shows DI distributions in the row versus iTimes format (traditionally used for RA tracking) for the overlapping region of the UV-2 and

Vis detectors. The row anomaly is a complex phenomenon that may result in artificially low or high values of reflectances, additionally affecting their wavelength dependence. The RA stems from an interplay of multiple factors that may affect the DI values. The RA is likely linked to a gradual detachment of the thermal blanket partially blocking some FOVs (rows). Since this blanket is highly reflective, its warped surface causes occasional (solar-angle dependent, predominantly affecting northern portion of the OMI orbit) reflection of the direct sunlight into some RA-affected rows. In addition, the reflective

surface leads to enhanced spatial cross-talk between adjacent RA-affected scenes (an anomalous stray light that is regulated by the wavelength- and angle- dependent reflectivity of the blanket). The time-, space- and wavelength-dependent combination of 3 factors may lead to increasing or decreasing DIs. Even more complications stem from the fact that the RA may increase inhomogeneity of the spectral-slit illumination, thus causing substantial (and unaccounted for) wavelength shifts and ensuing spectral distortions in the reflectances.


Deciphering the complex RA-related patterns in Figure 11, we first of all relate them to the pre-RA epoch that shows increase in the above-threshold cases in the equatorial regions, with a pronounced minimum centered on the sunglint domain (rows 10-30 and iTimes~ 650-1000). At the same time, the numbers of the above-threshold cases diminish towards the higher latitudes and the OMI's swath edges. In the TOA reflectances coming from the sunglint areas, the higher proportion

of the directly reflected sun light leads to a higher radiance-irradiance correlation, thus lower DIs. The latitudinal and swath-angle dependent trends can be linked to the gradually diminished influence of the surface that modulates the TOA reflectances due to the wavelength-dependent surface albedos. In the planned upgrade of the DI algorithm, we intend to address this component, thus decreasing the impact of geophysical factors.

Turning our attention to the RA-affected areas in Figure 11, we notice that DIs closely delineate the RA-affected areas and show pronounced north-south asymmetry, with intricate patterns of the relatively higher/lower DIs compared to the RA-free plots. The north-south asymmetry is caused by the well-documented northward growth (Schenkeveld et al. 2017) of the blanket-reflected direct-sunlight component in the RA-affected radiances. This inevitably lessens the corresponding DIs. The solar influence appears to be strongly cross-swath modulated. This is a new aspect that requires a detailed follow-up study

that is beyond the scope of the paper.  At the same time, the remainder of the RA-affected areas show significant increase in the above-threshold DIs.  This likely comes from  the RA-imposed and unaccounted for wavelength shifts.

The lower DI counts in the sunglint areas in Figure 11 seemingly contrast with the increased DI values at the center of these regions in the Vis (Figure 10). One should note that Figures 10 and 11 sample different spectral domains, with the 445.3-

455.7 nm range (Figure 10) known to be highly susceptible to saturation, contrary to the exceedingly rare incidence of saturation in the 349.9-360.3 nm band (Figure 11).

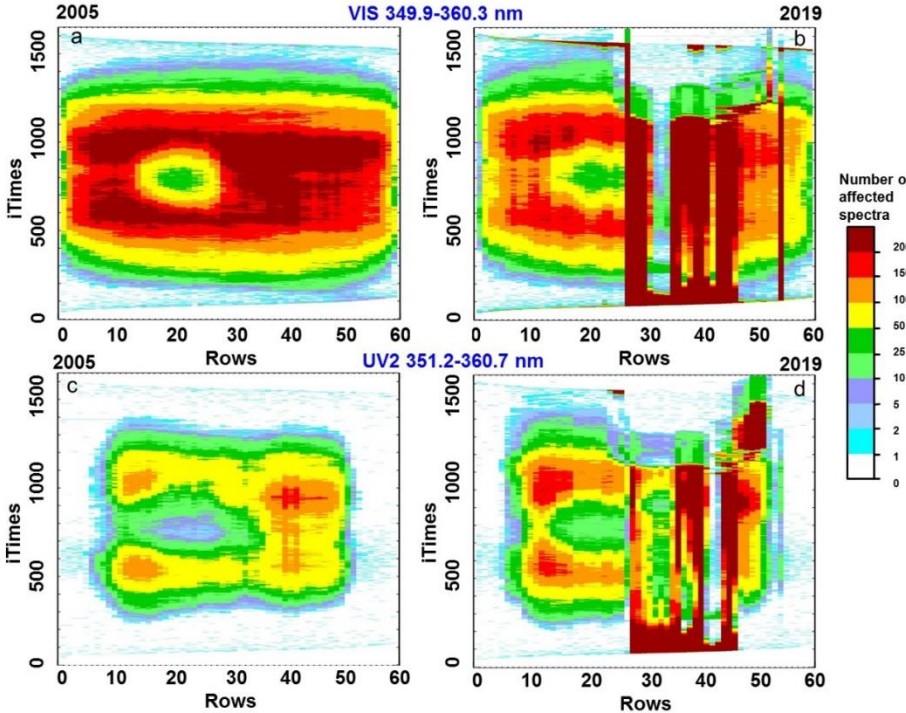

**Figure 11: Distribution of the number of affected spectra (DI>0.01) for March 2005 (left) and 2019 (right) for the Vis 349.9-360.3 nm (top) and UV2 351.2-360.7 nm (bottom). The row anomaly is responsible for the stripes of high values shown in 2019.**


### 4 Discussion and Conclusions

The OMI PixelQualityFlags (PQF) were designed to characterize each wavelength of the OMI spectrum (SPW flag is just one of the 16 bits in the PQF). The DI, developed on the basis of the correlations between observed and solar spectra, can serve as a simple but effective and complementary method for detecting and discarding anomalous UV and Vis satellite

spectra, for example associated with detector saturation, blooming, charge transfer or readout, excessive noise, cases of very low reflectance such as solar eclipse, or the OMI row anomaly. The DI summarizes all changes in the spectrum in one parameter and eliminates the need to examine all the available flags for a given pixel. An important motivation for introducing such an index is the convenience of handling it. For example, to infer enhanced information of the quality of spectra in the Vis region, we introduce 14 scalar-valued DIs for regions of the spectrum. For comparison, there are 751

binary saturation flags per spectrum in the level 1b. Similarly, we use 6 DIs for the UV2 spectrum; much less than the 577 flags assigned in the level 1b. Interpreting a large number of flags can be difficult. The DI product gives an indication of spectral quality based on overall correlation that is easier to interpret. Assessment of the DI the OMI Collection 3 L1b record has motivated improvements in detector corrections for the next version of the L1b product to be released in OMI Collection 4. The continuous nature of the DI allows data users to assign lower confidence to regions of the spectra that may not be

completely saturated as detected by an electronic saturation algorithm. DI values vary for spectra that do not experience any anomalies. These variations of the DI may carry information that can be used for other purposes. For instance, the DI can be used to search for areas of clear ocean water, in which the spectra are not abnormal, but experience significant deviations from the solar spectrum due to geophysical reasons.

The DI can be a useful tool for analyzing spectra obtained from other current and future space-borne sensors that may suffer from saturation and blooming such as TROPOMI (launched in 2017) or the similar Environmental trace gases Monitoring Instrument (EMI) on the GaoFen-5 satellite (Chen, 2016) (launched in 2018). Similar sensors include the OCO-2 (launched in 2014) and OCO-3 (launched in 2019) (Eldering et al., 2019), South Korea's geostationary Environment Monitoring Spectrometer (GEMS) (launched Feb. 18, 2020), NASA's geostationary Tropospheric Emissions: Monitoring of Pollution

(TEMPO) (Zoogman et al., 2017) (planned for launch in 2022), the Copernicus geostationary Sentinel-4 (planned for launch in 2023) and low-Earth orbit Sentinel-5 (planned for launch in 2023). Many of these sensors have a smaller pixel size and/or smaller field of view (FOV) than OMI. For such instruments, this may lead to an increase in the effects of sun glint. Studies utilizing the DI with current instruments may benefit the design of future instruments by identifying how often and under what conditions spectra are impacted by non-linear effects.

**Data availability**

The Decorrelation Index data for OMI Collection 3 data will be available at NASA Goddard Earth Sciences Data and Information Services Center (GES DISC). The OMI Level 1b data used for calculations of the DI are available at https://aura.gesdisc.eosdis.nasa.gov/data/Aura_OMI_Level1/. MODIS data are available at https://worldview.earthdata.nasa.gov/.

**Competing interests**

The authors declare that they have no conflicts of interest.

**Authors contributions**

NG developed computer codes, analyzed the DI results, and wrote the manuscript. ZF supported the development and implementation of the algorithms and comparison DI results with the ocean reflectance. DH proposed a concept of DI and wrote the manuscript. SM proposed a concept of DI and supported the development and implementation of the algorithms. JJ

set the task of developing an effective method for determining solar glints, supported the development of the algorithm, and wrote the manuscript. AV supported the development of the algorithm, and wrote the manuscript.

**Financial support**

This work was supported by the NASA Aura project (OMI core team) managed by Ken Jucks.

**Acknowledgements**

The authors thank the OMI and MODIS instrument teams for providing the OMI and MODIS data presented, respectively. We dedicate this work to the memory of Remco Braak, whose early work on saturation in OMI spectra helped to motivate this work.

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
