# Peer review of "Detection of anomalies in the UV/Vis reflectances from the Ozone Monitoring Instrument"

_Atmospheric Measurement Techniques, 2020_

## Referee Comment (RC1) · Anonymous Referee #1 · 22 Oct 2020

General Comments:

The paper describes a threshold method to detect anomalies in Level 1 radiance data, that may be difficult to identify in instrument radiance calibration, such as detector blooming. Flagging thresholds are given for the OMI instrument, and the authors claim that the method is applicable, with probably different thresholds, to other instruments as well. The paper is suitable for publication in AMT, as it shows a way to improve the flagging of suspicious radiance data in (satellite) spectrometers.

However, I am missing some important pieces of information:

1) how to calculate the Decorrelation Index DI exactly
2) a systematic discussion of what influences DI and why

3) how were the threshold values for DI in each spectral interval derived?

The first point could easily be mended by providing a mathematical formula for DI.

The second point is treated dispersed through the paper, with some information only anecdotically. The authors state that aerosols, Rayleigh scattering, blue water give rise to changes / increases in DI, and provide a few examples, without explicitly telling why DI is influenced. Without providing an exact calculation method of DI that is difficult to understand at first. Assuming that "r" is the Pearson's correlation coefficient, the slope of the spectrum also enters the equation. However, that also implies that the threshold reacts different to anomalies for ascending and descending spectral slopes. This should be discussed at the beginning, after introducing "r". Also it should be mentioned explicitly that perfectly normal spectral features, such as atmospheric absorption lines, give rise to enhanced DI. Several other aspects on the behaviour of DI are stated but may require reading between the lines to understand; in general statements should explicitly explained and without the need to reread sentences several times. Insight in the meaning of DI would be much clearer if all influences were discussed in one place, e.g. Section 2.2, instead of providing something in the introduction and letting others develop along the line as examples are given.

It remains intransparent how threshold values for DI were derived. But that is important if the reader wishes to apply the method to other instruments. Were thresholds derived based on radiative transfer calculations (including Raman scattering, aerosols, generic spectral surface albedos?) or was this a trial-and-error process until some credible results were obtained?? Was there any deliberate matching to OMI saturation flags?

Specific Comments:

Title (and abstract): the use of the word "non-linear" is not appropriate if also straylight is included (assumed is spatial straylight??). It may be that straylight has a non-linear

effect on calibrated radiances compared to TOA, but for the instrument the amount of straylight is linearly proportional to the amount of input light into the telescope (for a certain geometry). The same for obstruction due to MLI. Also a cosmic hit may (statistically) be linear with particle flux (for a given particle type/energy/angle). It would be more precise to use the word "anomaly". (in line 73 this is correctly used for OMI flags)

Line 35: I understand this introduction is the standard advertising for OMI, but OMI NRT/VFD dissimination is really irrelevant for this paper - please remove.

Line 48: it is not clear that the term "blooming" is explained by the first part of the sentence. This is then better done in line 65. Please move and integrate line 65 to here.

Line 54: this may suggest all GOME-2 (2A, 2B, 2C) sensors have an issue with clouds. As far as I recollect the issue was solved by introducing coadding. Please rephrase or leave this out, since you make abundantly clear that saturation effects are common. Maybe mention that saturation is simply a common effect due to the much larger dynamic range of TOA radiance compared to detector dynamic range.

Line 80-86: this is a general statement on radiance versus irradiance. While it is OK to make such a statement in the introduction, it is not sufficient to regard this as explanation of the behaviour of DI (see general comment).

Line 83-85: ...depends mainly on the strength(depth) of solar Fraunhofer features... The depth of the solar lines by itself doesn't change DI. What you want to say is that the low radiance in line cores makes them more susceptible to additive effects. Please rephrase. I wonder if the sentence is not better moved to a section that describes DI more in detail (see general comment).

Line 116-119: a mathematical formula for the calculation of DI should be given here.

Line 124: is DI not always >= 0 if atmospheric absorption is present ? (formula for

DI needed !) and what is the influence of a non-flat spectral albedo on DI ? (again: formula for DI needed). In general, would DI not always be >= 0 unless the reflectivity decreases with wavelength? As per my general comment, I propose to discuss that here, together with noise effects, and the resulting behaviour on parameters currently discussed in lines 80-86.

Table 1: the comment "strong spectral lines" is unclear. And why does this coincide with low DI thresholds?

Table 1, DI thresholds: see my general comment 3). Even if DI thresholds may depend on application, it must be described how the thresholds in this paper were derived, such that users may get a handle on how to set the threshold (for their application or for other instruments). "We just take these values and it works" is not enough. IMPORTANT: The paper is not acceptable without a proper description here. Scientific results must be reproducible and traceable. (as you know of course... I don't expect pages with analysis but say what you did so others can replicate)

Line 269-280: this is one of those examples where it is left to the reader to guess why exactly DI is deviant. The basics of this (spectral slope ?) should have been laid out before (see comment to line 124) and it would not harm to remind here why Rayleigh scattering has an effect ("contributes significantly to the top-of-atmosphere radiance" is a bit non-descript..)

Line 302: Why does scattered light from the thermal blanket "leads to the significant decrease of DIs". Not increase? And does "the blocking of the incoming Earth shine" result in distortion of the spectral shape ??

Line 321: why does low reflectivity (solar eclipse) increase DI but scattered light (line 301-303) lowers DI (should that also be: increases)? Is the solar eclipse effect due to noise or due to spatial straylight from around the occultation zone? Is scattered light not spatial straylight?

[Figure]

Line 329-332: I wonder if a high-pass filtering (e.g. dividing radiance by a local polyno-mial fit) would not largely remove the effects of aerosol and surface reflectivity on DI, and provide better sensitivity to anomalies in Vis. Also the "search for areas of clear ocean water" could probably just as well be done using a "slope index" based on 2 (continuum) wavelengths. Please comment / address.

Typographical comments:

The figures provide important visual information. However, most are disproportionally large compared to the text and to the required resolution (it is really not necessary to visually locate every single pixel). Especially figures [2,] 3,5,6,[8,9,] 10,11 should be reduced in size.

Also the font size in Table 1 is disproportionally large and should be reduced.

Typos:

double dots in lines 51, 94

double white space in line 169 ?

Dis -> DIs (?) line 301, 303

missing space line 303 (Fig. 11.Figure 11)

(otherwise kudos for a well-edited syntax !)

---

## Referee Comment (RC2) · Anonymous Referee #2 · 23 Oct 2020

General comments

The paper "Detection of non-linear effects in satellite UV/Vis reflectance spectra: Application to the Ozone Monitoring Instrument" by Nick Gorkavyi et al addresses the satellite radiance measurements and effects which may degrade the accuracy of these measurements. The authors propose a new method for detection of suspect or erroneous data, which is based on a correlation coefficient between the observed Earthshine radiance and sollar irradiance spectra.

The paper is structured as follows. After Introduction, in Section 2 the authors describe data and methods. This includes description of the Ozone Monitoring Instrument (OMI)

[Figure]

in subsection 2.1 and the definition of the new parameter – the Decorrelation Index (DI) which is zero in case of the perfect correlation between the Earth radiance and the irradiance spectra and increases with with increasing deviations between the two. In Section 3 the authors compare detected values of DI with the number of SPW flags for various "problematic areas" like cloud-covered regions, lakes and oceans. Further, orbital and global distribution of the number of affected spectra are reported. The effect of the so-called row anomaly connected to instrumental issues is discussed in Section 3.4. In the discussion Section potential usefulness of the DI method is discussed and possible perspective studies utilizing DI for other space-borne sensors is mentioned.

The paper is clearly written and the subject of the manuscript is of relevance for atmospheric studies, as inaccurate measurements impact retrieval of atmospheric constituents including trace gases (like ozone) or aerosols. The method proposed in the manuscript should be advantageous over existing error detectors (Saturation_Possibility_Warning - SPW Flags) as it is sensitive to any distorsions of the reflectance spectrum, regardless of its cause and provides a range of deviations, not just a binary output. In my opinion the manuscript is suitable for publication in AMT after some issues are clarified.

Specific comments

1. It is not clear to me how in practice the DI coefficient was calculated and how the threshold values for different wavenumber ranges given in Table 1 were established.

2. In figure 1 authors compare DI with the number of SPW flags for a very restricted range in the spectral space (414-424nm). It is not clear why such range was chosen – it is different in Figs. 2 and 3.

3. According to table 1 on page 5, DI thresholds for damaged spectra depend on the spectral region and vary considerably (by two orders of magnitude). On the other hand, in figures 2,3,5,6 only the actual value of DI is plotted. It is therefore difficult to say how much DI exceeds the threshold. I suppose it would be better to divide the actual value

of DI by the threshold value for the particular spectral range to better illustrate the degree of deviations.

4. In the introduction the authors address two different effects which may deteriorate measurement data: saturation and blooming. After reading description on page 2 it is not clear to me how to differentiate in practice between the effect of the two. In both cases, as the authors write, flow of excessive electrons to neighboring pixels occurs.

Technical corrections:

R1: the shortcut OMI is first used in line 20 but introduced later in line 22

R2: Shortcut CCD is first used in line 46 but introduced later in line 66

R3: line 94 "orbit orbit...13:45.." the word is written twice and there is a double dot at the end of the sentence

---

## Author Comment (AC1) · 6 Dec 2020

We thank the reviewer for their time and effort reviewing this manuscript and for providing helpful and constructive comments that have helped to improve the manuscript.

RC1: "However, I am missing some important pieces of information:

1) how to calculate the Decorrelation Index DI exactly

2) a systematic discussion of what influences DI and why

3) how were the threshold values for DI in each spectral interval derived?

[Figure]

The first point could easily be mended by providing a mathematical formula for DI. The second point is treated dispersed through the paper, with some information only anecdotically. The authors state that aerosols, Rayleigh scattering, blue water give rise to changes / increases in DI, and provide a few examples, without explicitly telling why DI is influenced. Without providing an exact calculation method of DI that is difficult to understand at first. Assuming that "r" is the Pearson's correlation coefficient, the slope of the spectrum also enters the equation. However, that also implies that the threshold reacts different to anomalies for ascending and descending spectral slopes. This should be discussed at the beginning, after introducing "r". Also it should be mentioned explicitly that perfectly normal spectral features, such as atmospheric absorption lines, give rise to enhanced DI. Several other aspects on the behaviour of DI are stated but may require reading between the lines to understand; in general statements should explicitly explained and without the need to reread sentences several times. Insight in the meaning of DI would be much clearer if all influences were discussed in one place, e.g. Section 2.2, instead of providing something in the introduction and letting others develop along the line as examples are given. It remains intransparent how threshold values for DI were derived. But that is important if the reader wishes to apply the method to other instruments. Were thresholds derived based on radiative transfer calculations (including Raman scattering, aerosols, generic spectral surface albedos?) or was this a trial-and-error process until some credible results were obtained?? Was there any deliberate matching to OMI saturation flags?"

AC1: We provide the requested equations in the revised section 2.2 and follow-up with an extensive discussion of the expected DI behaviour:

"Evidently DI=0 for the simple case of a perfect match with I=const*F_0; if I=-const*F_0, then DI=2. If I and F_0 are completely unrelated, then DI=1. Considering the 'smooth' (low-frequency) component of I and F_0, we expect them to be generally correlated in the spectral regions relatively free of major atmospheric absorptions (ozone in particular). The correlation would be inevitably diminished by the wavelength-dependent

Rayleigh scattering and surface reflectivity. Once a multitude of deep spectral lines is superimposed on a smooth envelope, DI will depend mainly on a match between the shape and position of these I and $F_0$ spectral transitions, with the correlation depending on the S/N of the tested radiances and irradiances, and even more so on slight (in OMI's case) wavelength mis-alignments between radiances and irradiances, with the steep line flanks magnifying the differences. An additional de-correlating factor is brought forth by the omni-present atmospheric rotational Raman scattering (e.g., Joiner et al., 1995). Under the circumstances, one may never expect DI=0 save the exceedingly rare cases of a perfect solar glint. It is known that Pearson's correlation coefficient is sensitive to outliers, thus simplifying detection of spectral distortions in the high-resolution data compared to the low-resolution cases, with the latter tending to lessen the impact of additive components (the shallower lines are potentially less susceptible to stray light), as well as the wavelength mis-alignment (spectral blending of multiple features leading to partial canceling of distortions in the adjacent features). At a given spectral resolution and S/N, DI sensitivity may grow with increasing numbers and contrasts (depths) of spectral features in the chosen spectral window. At the same time, the DI is expected to be sensitive to artifacts associated with cosmic ray hits. The interval 440-480 nm, where there are few deep spectral lines, should be especially sensitive to geophysical factors, for example, to the wavelength-dependent albedo of the earth's surface. Note that there are cases when direct solar radiation $F_0$ is mixed with due to instrument problems (see below). Under this specific circumstance DI will decrease, since correlation between $(I+dF_0)$ and $F_0$ is always higher (thus DI lower) than between I and $F_0$. A similar effect occurs with sunglint from the water surface, when the proportion of directly reflected sunlight in I increases significantly. Note that in the current approach we do not compensate for the relatively smooth spectral differences imposed by atmospheric (Rayleigh scattering) and surface (wavelength-dependent albedo) factors, leaving this to the next DI version. This step would make DI more sensitive to the instrument-imposed anomalies, further disentangling those from the geophysical factors (see below)".

We return to this discussion on multiple occasions (the blooming effects in particular) throughout the revised text. We also provide the explanation about the proposed thresholds:

"The provisional (the user may redefine the values using the auxiliary data provided in the OMI DI product) DI thresholds were determined as follows. We used all available, mission-long OMI UV2 and Vis radiances. For each orbit and for every spectral window we constructed DI histograms. Then we selected numerous cases sampling the tails of the DI histograms. On a case-by-case basis, for different scenes and spectral windows, we found empirically the lowest-DI values that repeatedly separate the scenes with apparently normal (spectrally smooth, with the fine-structure, low-amplitude Raman-scattering features) and distorted reflectances. These DI thresholds approximately correspond to 99.995-99.998 percentiles in the DI distributions. We plan to provide a statistically rigorous threshold definition in the improved DI version". We have not deliberately matched the operational and the DI flags, though comparing them on multiple occasions (see Figures 1-7).

RC2: "Specific Comments: Title (and abstract): the use of the word "non-linear" is not appropriate if also straylight is included (assumed is spatial straylight??). It may be that straylight has a non-linear effect on calibrated radiances compared to TOA, but for the instrument the amount of straylight is linearly proportional to the amount of input light into the telescope (for a certain geometry). The same for obstruction due to MLI. Also a cosmic hit may (statistically) be linear with particle flux (for a given particle type/energy/angle). It would be more precise to use the word "anomaly". (in line 73 this is correctly used for OMI flags)"

AC2: New title: Detection of anomalies in the UV/Vis reflectances from the Ozone Monitoring Instrument

RC3: "Line 35: I understand this introduction is the standard advertising for OMI, but OMI NRT/VFD dissimination is really irrelevant for this paper - please remove."

AC3: The reference to the OMI NRT/VFD dissemination has been removed.

RC4: "Line 48: it is not clear that the term "blooming" is explained by the first part of the sentence. This is then better done in line 65. Please move and integrate line 65 to here."

AC4: The explanation of the blooming effect has been moved to where it was first mentioned.

RC5: "Line 54: this may suggest all GOME-2 (2A, 2B, 2C) sensors have an issue with clouds. As far as I recollect the issue was solved by introducing coadding. Please rephrase or leave this out, since you make abundantly clear that saturation effects are common. Maybe mention that saturation is simply a common effect due to the much larger dynamic range of TOA radiance compared to detector dynamic range."

AC5: The reference to GOME-2 is removed.

RC6: "Line 80-86: this is a general statement on radiance versus irradiance. While it is OK to make such a statement in the introduction, it is not sufficient to regard this as explanation of the behaviour of DI (see general comment)."

AC6: A systematic discussion of the factors that affect DI is provided in Section 2.2 that has been substantially revised.

RC7: "Line 83-85: ...depends mainly on the strength(depth) of solar Fraunhofer features... The depth of the solar lines by itself doesn't change DI. What you want to say is that the low radiance in line cores makes them more susceptible to additive effects. Please rephrase. I wonder if the sentence is not better moved to a section that describes DI more in detail (see general comment)."

AC7: We amend a part of Introduction (there we talk about the 'traditional' correlation) and return to the subject in the revised Sect. 2.2.

RC8: "Line 116-119: a mathematical formula for the calculation of DI should be given

here."

AC8: The equation for the DI is given in the revised Sect. 2.2.

RC9: "Line 124: is DI not always >= 0 if atmospheric absorption is present? (formula for DI needed !) and what is the influence of a non-?at spectral albedo on DI? (again: formula for DI needed). In general, would DI not always be >= 0 unless the reflectivity decreases with wavelength? As per my general comment, I propose to discuss that here, together with noise effects, and the resulting behavior on parameters currently discussed in lines 80-86."

AC9: Please see the revised sect. 2.2, as well as additional comments throughout the revised text.

RC10: "Table 1: the comment "strong spectral lines" is unclear. And why does this coincide with low DI thresholds?"

AC10: We provide additional details in the revised Sect. 2.2.

RC11: "Table 1, DI thresholds: see my general comment 3). Even if DI thresholds may depend on application, it must be described how the thresholds in this paper were derived, such that users may get a handle on how to set the threshold (for their application or for other instruments). "We just take these values and it works" is not enough. IMPORTANT: The paper is not acceptable without a proper description here. Scientific results must be reproducible and traceable. (as you know of course... I don't expect pages with analysis but say what you did so others can replicate)"

AC11: Please see the answer to RC1 who had a similar concern. We have added relevant clarifications to the text of the article to address these points.

RC12: "Line 269-280: this is one of those examples where it is left to the reader to guess why exactly DI is deviant. The basics of this (spectral slope?) should have been laid out before (see comment to line 124) and it would not harm to remind here why Rayleigh scattering has an effect ("contributes significantly to the top-of-atmosphere

radiance" is a bit non-descript..)"

AC12: We amend the text in the discussion of Sect. 3.3 to address this point.

RC13: "Line 302: Why does scattered light from the thermal blanket "leads to the significant decrease of DIs". Not increase? And does "the blocking of the incoming Earth shine" result in distortion of the spectral shape ??"

AC13: We expand and clarify the discussion of the RA phenomenon and link it to the observed DI patterns.

RC14: "Line 321: why does low reflectivity (solar eclipse) increase DI but scattered light (line 301-303) lowers DI (should that also be: increases)? Is the solar eclipse effect due to noise or due to spatial straylight from around the occultation zone? Is scattered light not spatial straylight?"

AC14: We elaborate on both questions in the revised text: "...a solar eclipse zone. Though we cannot disentangle all the contributing factors for the latter, here we mention two of them as the likely causes of the high DI values (thus enhanced distortions in the reflectances): the low S/N of the eclipsed radiances, as well as the drastically increased portion (compared to the normally-lit scenes) of the additive (straylight) component." "...the reflective surface leads to enhanced spatial cross-talk between adjacent RA-affected scenes (an anomalous stray light that is regulated by the wavelength- and angle-dependent reflectivity of the blanket)."

RC15: "Line 329-332: I wonder if a high-pass filtering (e.g. dividing radiance by a local polynomial fit) would not largely remove the effects of aerosol and surface reflectivity on DI, and provide better sensitivity to anomalies in Vis. Also the "search for areas of clear ocean water" could probably just as well be done using a "slope index" based on 2 (continuum) wavelengths. Please comment/address."

AC15: We plan to develop a revised index DI-2 in the future, which will use the decomposition of the spectrum into high-frequency and low-frequency components; this

should decrease the impact of geophysical factors. We mentioned this in the revised text. .

RC16: "Typographical comments: The figures provide important visual information. However, most are disproportionally large compared to the text and to the required resolution (it is really not necessary to visually locate every single pixel). Especially figures [2,] 3,5,6,[8,9,] 10,11 should be reduced in size".

AC16: The sizes of all the listed figures are reduced

RC17: "Also the font size in Table 1 is disproportionally large and should be reduced."

AC17: The font in the Table 1 is reduced

RC18: "Typos: double dots in lines 51, 94 double white space in line 169 ? Dis -> DIs (?) line 301, 303 missing space line 303 (Fig. 11.Figure 11) (otherwise kudos for a well-edited syntax !)"

AC18: All typos have been corrected.

---

## Author Comment (AC2) · 6 Dec 2020

We thank the reviewer for their time and effort reviewing this manuscript and for providing helpful and constructive comments that have helped to improve the manuscript.

RC1: "It is not clear to me how in practice the DI coefficient was calculated and how the threshold values for different wavenumber ranges given in Table 1 were established."

AC1: The equation for the Decorrelation Index (DI) is written in 2.2. The DI is a mathematically strictly calculated parameter, and the threshold is only a rough estimate. We have added relevant clarifications to the text of the article:

[Figure]

"The provisional (the user may redefine the values using the auxiliary data provided in the OMI DI product) DI thresholds were determined as follows. We used all available, mission-long OMI UV2 and Vis radiances. For each orbit and for every spectral window we constructed DI histograms. Then we selected numerous cases sampling the tails of the DI histograms. On a case-by-case basis, for different scenes and spectral windows, we found empirically the lowest-DI values that repeatedly separate the scenes with apparently normal (spectrally smooth, with the fine-structure, low-amplitude Raman-scattering features) and distorted reflectances. These DI thresholds approximately correspond to 99.995-99.998 percentiles in the DI distributions. We plan to provide a statistically rigorous threshold definition in the improved DI version".

RC2: "In figure 1 authors compare DI with the number of SPW flags for a very restricted range in the spectral space (414-424nm). It is not clear why such range was chosen – it is different in Figs. 2 and 3."

AC2: DI for 14 spectral intervals of ∼10 nm are calculated in the Vis range of 350-498 nanometers. The behavior of the DI in each interval has its own characteristics. In Fig. 1, we showed the spatial dependence of the DI for one interval, which has a significant sensitivity to changes in the spectrum, in Fig. 2 - spectral dependence of the DI for several intervals, in Fig. 3 - spectral dependence of the DI for the entire Vis area.

RC3: "According to table 1 on page 5, DI thresholds for damaged spectra depend on the spectral region and vary considerably (by two orders of magnitude). On the other hand, in figures 2,3,5,6 only the actual value of DI is plotted. It is therefore difficult to say how much DI exceeds the threshold. I suppose it would be better to divide the actual value of DI by the threshold value for the particular spectral range to better illustrate the degree of deviations."

AC3: Figures 2,3,5,6 show real DI, which mathematically strictly show the correlation between the terrestrial and solar spectrum. The thresholds that are set in Table 2 are very rough estimates. If we divide the exact result by an approximate factor, which

varies greatly from interval to interval, we make it difficult to quantitatively interpret the picture.

RC4: "In the introduction the authors address two different effects which may deteriorate measurement data: saturation and blooming. After reading description on page 2 it is not clear to me how to differentiate in practice between the effect of the two. In both cases, as the authors write, flow of excessive electrons to neighboring pixels occurs."

AC4: An improved version of the explanation for both effects is now as follows:

"Saturation occurs when bright light causes the number of electrons in a sensor pixel to exceed either the maximum charge capacity of an individual charge-coupled device (CCD) photodiode, or the maximum charge transfer capacity of the sensor. A blooming effect occurs when electrons from a highly illuminated pixel of the CCD matrix jump to a neighboring pixel, causing distortion of its signal."

Technical corrections: RC5: "the shortcut OMI is first used in line 20 but introduced later in line 22"

AC5: The introducing the shortcut OMI has been moved to where it was first mentioned.

RC6: "Shortcut CCD is first used in line 46 but introduced later in line 66"

AC6: The introducing the shortcut CCD has been moved to where it was first mentioned.

RC7: "line 94 "orbit orbit...13:45.." the word is written twice and there is a double dot at the end of the sentence"

AC7: All typos have been corrected.